# Association of miR-499 Polymorphism and Its Regulatory Networks with Hashimoto Thyroiditis Susceptibility: A Population-Based Case-Control Study

**DOI:** 10.3390/ijms221810094

**Published:** 2021-09-18

**Authors:** Farhad Tabasi, Vahed Hasanpour, Shamim Sarhadi, Mahmoud Ali Kaykhaei, Pouria Pourzand, Mehrdad Heravi, Ahmad Alinaghi Langari, Gholamreza Bahari, Mohsen Taheri, Mohammad Hashemi, Saeid Ghavami

**Affiliations:** 1Department of Clinical Biochemistry, School of Medicine, Zahedan University of Medical Sciences, Zahedan 9816743463, Iran; f.tabasi@modares.ac.ir (F.T.); p.poorzand1@gmail.com (P.P.); mehrdadh3731@gmail.com (M.H.); ghrb1333@gmail.com (G.B.); 2Department of Physiology, Faculty of Medical Sciences, Tarbiat Modares University, Tehran 1411713116, Iran; 3Student Research Committee, School of Medicine, Zahedan University of Medical Sciences, Zahedan 9816743463, Iran; vahedhasanpour@gmail.com; 4Department of Medical Biotechnology, Faculty of Advanced Medical Sciences, Tabriz University of Medical Sciences, Tabriz 5166616471, Iran; shamim.sarhadi65@gmail.com; 5Genetics of Non-Communicable Disease Research Center, Zahedan University of Medical Sciences, Zahedan 9816743463, Iran; mazyar44@gmail.com (M.A.K.); amirt112@yahoo.com (M.T.); 6Department of Endocrinology, School of Medicine, Zahedan University of Medical Sciences, Zahedan 9816743463, Iran; 7Student Research Committee, School of Medicine, Kerman University of Medical Sciences, Kerman 7616913555, Iran; a.alinaghi.langari@gmail.com; 8Children and Adolescent Health Research Center, Resistant Tuberculosis Institute, Zahedan University of Medical Sciences, Zahedan 9816743463, Iran; 9Research Institute of Oncology and Hematology, Cancer Care Manitoba, University of Manitoba, Winnipeg, MB R3E 0V9, Canada; 10Biology of Breathing Theme, Children Hospital Research Institute of Manitoba, University of Manitoba, Winnipeg, MB R3E 0V9, Canada; 11Faculty of Medicine, Katowice School of Technology, 40-555 Katowice, Poland; 12Department of Human Anatomy and Cell Science, Rady Faculty of Health Sciences, Max Rady College of Medicine, University of Manitoba, Winnipeg, MB R3E 0J9, Canada

**Keywords:** Hashimoto thyroiditis, autoimmune thyroid disease (AITD), mir-499, single nucleotide polymorphism (SNP), regulatory network

## Abstract

Hashimoto thyroiditis (HT) is a common autoimmune disorder with a strong genetic background. Several genetic factors have been suggested, yet numerous genetic contributors remain to be fully understood in HT pathogenesis. MicroRNAs (miRs) are gene expression regulators critically involved in biological processes, of which polymorphisms can alter their function, leading to pathologic conditions, including autoimmune diseases. We examined whether miR-499 rs3746444 polymorphism is associated with susceptibility to HT in an Iranian subpopulation. Furthermore, we investigated the potential interacting regulatory network of the miR-499. This case-control study included 150 HT patients and 152 healthy subjects. Genotyping of rs3746444 was performed by the PCR-RFLP method. Also, target genomic sites of the polymorphism were predicted using bioinformatics. Our results showed that miR-499 rs3746444 was positively associated with HT risk in heterozygous (OR = 3.32, 95%CI = 2.00–5.53, *p* < 0.001, CT vs. TT), homozygous (OR = 2.81, 95%CI = 1.30–6.10, *p* = 0.014, CC vs. TT), dominant (OR = 3.22, 95%CI = 1.97–5.25, *p* < 0.001, CT + CC vs. TT), overdominant (OR = 2.57, 95%CI = 1.62–4.09, *p* < 0.001, CC + TT vs. CT), and allelic (OR = 1.92, 95%CI = 1.37–2.69, *p* < 0.001, C vs. T) models. Mapping predicted target genes of miR-499 on tissue-specific-, co-expression-, and miR-TF networks indicated that main hub-driver nodes are implicated in regulating immune system functions, including immunorecognition and complement activity. We demonstrated that miR-499 rs3746444 is linked to HT susceptibility in our population. However, predicted regulatory networks revealed that this polymorphism is contributing to the regulation of immune system pathways.

## 1. Introduction

Hashimoto thyroiditis (HT) is an organ-specific autoimmune disease and the most prevalent type of autoimmune thyroid disease (AITD), characterized by immunocyte imbalance, autoantibodies production, and increased expression and activity of inflammatory mediators [1]. With a prevalence of 4.5% (both clinical and subclinical cases) based on biochemical assessments, HT is the most common cause of spontaneous hypothyroidism in areas with sufficient iodine intake; of note, the prevalence of HT determined by cytological diagnosis is even higher and estimated to reach 13.4% [2]. Annually, 4% of cases with subclinical hypothyroidism develop overt hypothyroidism, a risk that increases with age [2]. The course, severity, and response to treatments of AITDs are variable among individuals [3], which may be related to variations in predisposing factors.

HT is characterized by epithelial cell destruction, diffuse infiltration of lymphocytic cells, fibrosis, and higher thyroid autoantibodies [2]. Although known as a multifactorial disease, accumulating evidence emphasizes the role of genetic predisposition in HT [4,5]. Several genes and their variants that may be linked with increased autoimmunity and HT risk, including genes that are regulating inflammatory and immunity processes, have been introduced [6,7]; however, given the multifactorial nature of HT and the complex genetic interactions underlying the disease, defining the exact etiological roles of these factors is challenging.

Micro RNAs (miRs) are small endogenous non-coding, single-stranded RNAs consisting of 18–23 nucleotides in length [8] that regulate gene expression at the post-transcriptional level by forming an RNA-induced silencing complex (RISC) and binding to the 3′ untranslated region (UTR) of the target mRNA, ultimately leading to degradation or inhibition of mRNA translation [9]. It has been estimated that miRs affect over 60% of human gene expression and are involved in the vast majority of physiological processes, including metabolism, development, immunity, and overall cell fate [9,10]. However, dysregulation of a miR can compromise its function and trigger a pathological situation. Several miRs have been implicated in immune response regulation associated with autoimmunity [11,12]. The *MIR499* is located on chromosome 20q11.22, within intron 20 of the beta-myosin heavy chain 7B (*MYH7B*), and encodes miR-499, which regulates several immune system functions through various mechanisms such as inflammatory cytokine signaling and production [13]. A potential contribution of miR-499 to autoimmunity and autoimmune diseases has been proposed [14].

Genetic variations can dramatically modify the expression and function of encoded gene products. Single-nucleotide polymorphisms (SNPs) constitute the most common type of variation in the human genome, and SNPs within miR genes can influence miR expression, maturation, and function, thereby affecting miR target expression [15]. Polymorphisms in pre-miR (a hairpin structure forms after transcription and turns into mature miR) may influence miR processing and maturation, or even secondary structure, affecting the function of target genes [16]. Rs3746444 has been identified as the polymorphism located at pre-miR-499 (position 73). Previous studies have suggested that rs3746444 can potentially alter the maturation and function of the corresponding miR [16]. Currently, little information regarding the connection between miR-499 rs3746444 polymorphism and AITDs is available [17]. Therefore, our study aimed to evaluate the possible associations between miR499 rs3746444 polymorphism and predisposition to HT in an Iranian study population. Additionally, we investigated the potential targets and interactions of miR-499 using online databases, bioinformatics, and computational biology.

## 2. Results

### 2.1. Genotypes, Allele Frequencies, and Their Association with HT Risk

This case-control study included 150 HT patients (17 males, 133 females) and 152 healthy subjects (24 males, 128 females). The mean age ± SD in the HT and control groups was 37.96 ± 12.06 and 36.22 ± 12.86, respectively. Sex and age were not statistically different between groups (*p* = 0.314 and *p* = 0.226; Table 1).

Genotypes and allele frequencies of miR-499 rs3746444 polymorphism are depicted in Table 2. The findings showed that the miR-499 rs3746444 polymorphism significantly increased the risk of HT in heterozygous (OR = 3.32, 95%CI = 2.00–5.53, *p* < 0.001, CT vs. TT), homozygous (OR = 2.81, 95%CI = 1.30–6.10, *p* = 0.014, CC vs. TT), dominant (OR = 3.22, 95%CI = 1.99–5.25, *p* < 0.001, CT + CC vs. TT), overdominant (OR = 2.57, 95%CI = 1.61–4.07, *p* < 0.001), and allele (OR = 1.92, 95%CI = 1.37–2.69, *p* < 0.001, C vs. T) genetic models (Table 2). Genotype distribution among controls for miR-499 rs3746444 (χ2 = 0.597, *p* = 0.440) showed no deviation from the HWE.

### 2.2. Predicted Gene Networks of miR-499

We used TargetScanHuman to predict target genes of miR-499. The top 100 predicted genes for the miR were then selected for network construction. Mapping predicted genes of tissue-specific protein–protein interaction-, co-expression-, and miR-TF networks are presented in Table 3 and Figure 1, Figure 2 and Figure 3—the complete list of each network’s statistics is provided in Appendix A. Table 3 represents nodes in each network with the highest degree and betweenness; a degree number describes connections to a node, and betweenness is a topological feature of a network representing a node centrality, defined by the number of times a node acts as a bridge along the shortest path between two other nodes and measures the shortest path fraction between node pairs through a certain node. Higher betweenness for a protein means most control on information flow by that protein in a given network [18]. Pathway enrichment of functional clusters of these networks by the Reactome biological pathway database is summarized in Table 4 and fully outlined in Appendix A. In network enrichment analysis performed in this study, total, expected, and hits are terms that represent the number of genes involved in analysis, the number of nodes in the constructed network which are expected to match with genes in functional gene sets, and the number of analyzed nodes matched with the gene sets used in functional enrichment analyses, respectively.

#### 2.2.1. Thyroid-Specific PPIN

The tissue-specific PPIN for thyroid demonstrated enrichment for pathways critically involved in immune system functions (Figure 1 and Table 4). This network showed a strong implication in the adaptive and innate immune system, antigen processing, and pattern recognition, mainly throughout toll-like receptors (TLRs) signaling transduction (Table 4). These results are consistent with previous findings suggesting that self-tolerance and antigen recognition are compromised in HT pathophysiology [19], in which the immune system reacts against self-antigens. Based on the network reconstruction results for thyroid-specific PPIN, we identified mesenchyme homeobox 2 (*MEOX2*), syndecan binding protein (*SDCBP*), ubiquitin-conjugating enzyme E2 D3 (*UBE2D3*), mitogen-activated protein kinase 8 (*MAPK8*), and nuclear factor kappa B inhibitor alpha (*NFKBIA*) as hub-driver nodes (Figure 1).

*MEOX2* is a transcription factor protein-coding gene of cell cycle inhibitors in endothelial cells, preventing vascular cell proliferation [20]. Notably, Meox2 is suggested to significantly serve as a negative regulator of gene expression, particularly in response to an inflammatory condition, and exerts its effects via IκBβ, which is a major isoform of IκB, and NF-κB inhibitor [21]. Therefore, Meox2, by interacting with crucial elements of the NF-κB pathway, can be considered a pivotal down regulator of NF-κB [22].

*SDCBP* encodes syntenin-1 (also known as melanoma differentiation-associated gene-9 (MDA-9)), an intracellular PDZ (Psd-95 (post synaptic density protein), DlgA (drosophila disc large tumor suppressor), and ZO1 (zonula occludens-1 protein)) containing protein, which is primarily involved in membrane-associated adherence junction and adhesion [23]. Notably, syntenin-1 interacts with numerous proteins, is implicated in trafficking and organization of proteins in the plasma membrane, and also in several immune system elements functions, including B-cell development [24], T-cell chemotaxis [25], interacting with pro-transforming growth factor-α [26] and interleukin (IL)-5 receptor α [27], and IL-5 signaling [28]. Moreover, syntenin physiologically suppresses TRAF6 and inhibits IL-1R/TLR4-mediated NF-κB activation pathways [29]. Eventually, syntenin-1 may play a role in the dynamic regulation of TLR7, which is involved in self-RNA recognition, and therefore autoimmunity [30].

*UBE2D3* is a protein-coding gene that encodes UbcH5c, a member of the E2 ubiquitin-conjugating enzyme family, participating in ubiquitination of multiple vital signaling pathways such as p53 tumor-suppressor [31] and NF-κB [32], an essential transcription factor for immune-related genes, and importantly, inflammation [33]. Therefore, UbcH5c via NF-κB and related pathways, regulating immune functions [32]. 

MAPK8 (also known as c-Jun *N*-terminal kinase 1 (JNK1)) belongs to the MAP kinase family and serves as an integration point for signaling pathways of numerous biological processes, mainly T-cell functions [34]. 

Finally, *NFKBIA* encodes IκBα (nuclear factor of kappa light polypeptide gene enhancer in B-cells inhibitor, alpha), a member of the NF-κB inhibitor family [35]. IκBα regulates NF-κB activity, which is crucial for many pro-inflammatory and immune responses [36,37,38]. Several polymorphisms in NF-κB inhibitors have been suggested in the pathophysiology of Graves’ disease [39].

To test our hypothesis and verify our results on enriched pathways regarding the predicted genes, we performed the post-analysis on genes with the highest degree (see Table 5 for network properties). The results of the STRING database illustrated a PPIN with at least six major clusters implicated pivotally in processes such as ubiquitination, protein catabolism, Wnt and NF-κB, and MAPK signaling cascade. Moreover, this network is implicated in immune system processes, including antigen processing, inflammation, and TLR signaling (Figure 4; further details are provided in Appendix A). Significantly, this network contributes to canonical Wnt and NF-κB regulation, critical for immune system activity [40,41,42,43]. Further, the local network cluster showed the canonical NF-kB pathway as the significance of the network (FDR = 0.0449). Thus, based on bioinformatics analyses, we argued that miR-499 is engaged in HT pathogenesis, mainly via Wnt and NF-κB pathways.

#### 2.2.2. Thyroid-Specific Co-Expression Genes Network

We also predicted the thyroid-specific sub-regulatory network for co-expression genes using the TCSBN database. Evaluation of the co-expression gene network related to regulatory modules of miR-499 revealed G protein-coupled receptor 65 (*GPR65*), T-cell activation RhoGTPase activating protein (*TAGAP*), myosin IF (*MYO1F*), mitogen-activated protein kinase kinase kinase kinase 1 (*MAP4K1*), and *CD5* as main driver nodes (Figure 2). This network is implicated in RhoGTPase signaling, immunoregulation between lymphoid and non-lymphoid cells, adaptive immune system function, IL receptor signaling, and B-cell receptor signaling (Table 4). Indeed, pathway enrichment of the network’s components showed a strong association with innate and adaptive immune system functions.

The protein encoded by *GPR65* (also known as T-cell death-associated gene 8 protein [TDAG8]) is a protein-sensing GPCR mainly expressed in lymphoid organs [44]. The GPR65 upregulates during programmed cell death of T lymphocytes [45] during thymocyte development, essential for self-tolerance [46], though, the GPR65 is not limited to T-cells and expresses in leukocytes and macrophages [47].

*TAGAP* encodes a protein (also known as ARHGAP47), expressed in immune cells, mainly in activated T-cells, and may have a role in T-cell activation and, therefore, immune regulation [48]. The altered expression in several human autoimmune disorders has been shown [49,50].

*MYO1F* is a protein-coding gene that encodes an unconventional type 1 myosin [51], expressed primarily in immune cells of mammals [52], and mainly involved in immune cell motility and adhesion, and also innate immunity [53].

*MAP4K1* encodes a Serine/threonine-protein kinase, known as hematopoietic progenitor kinase 1 (HPK1), involved in various signaling pathways, including B and C lymphocyte receptors and TGF-β [54,55,56,57]. The HPK1 links the T-cell receptor (TCR) stimulation and NF-κB activation [58] and also negatively regulates activator protein 1 (AP-1) [55]. Moreover, HPK1 may be a key regulator of T-cell survival [59].

Finally, *CD5* encodes a type-I transmembrane glycoprotein in all mature T-cells, thymocytes, and a small B-cell subpopulation [60] and negatively regulates TCR signaling and T-cell activation [61,62]. Thus, CD5 dysregulation may be associated with autoimmunity [63].

We performed post-analysis based on selected predicted genes from the co-expression network. The retrieved network constructed by the STRING database demonstrated that the PPIN from predicted genes contributed significantly to several immune system processes, mainly immune response, catalytic, and GTPase activity regulation (Figure 5. Further details are provided in Appendix A; see Table 5 for network properties). The local network cluster indicated its implication on BCR and TCR-related activity (FDR = 1.17 × 10^−5^ and 0.0091, respectively). These results suggest the role of predicted co-expression genes on immune response, notably mediated by lymphocytes, which involve cellular and humoral immunity.

#### 2.2.3. Thyroid-Specific miR-TF Network

We identified Zic family member 2 (*ZIC2*), NFKBIA, member of RAS oncogene family (*RAP2C*), homeobox A5 (*HOXA5*), and transcription factor 7 like 2 (*TCF7L2*) as prominent driver nodes of the miR-TF network (Figure 3). Functional regulatory cluster enrichment revealed that these nodes are mainly involved in immune system pathways, including TLR signaling and pro-inflammatory cytokine induction.

*ZIC2* encodes zinc, a member of the zinc finger protein family, with transcriptional activity. ZIC2 overexpression may be associated with apoptosis inhibition [64] and lymphocyte infiltration [65]. For details of *NFKBIA*, see the Section 2.2.1.

*RAP2C* encodes a protein that belongs to the Ras GTPase superfamily involved in cellular proliferation, differentiation, and apoptosis regulation. This protein is implicated in innate immunity and Akt signaling pathway [66]. Rap2 can enhance MAP4K-related JNK activation [67]. Additionally, Rap2c, via the MAPK pathway, promotes proliferation and inhibits apoptosis [68].

*HOXA5* encodes a member of homeobox transcription factors. Hox5a is involved in the expression of genes that regulate proliferation [69] and seems to be an essential element in myeloid differentiation [70].

Finally, *TCF7L2* encodes a T-cell specific high mobility group (HMG)-box transcriptional factor, known as T-cell factor 4 (TCF4). Dysregulated expression of TCF4 has been demonstrated in several human cancers [71,72] and implies autoimmune diabetes [73,74].

Post-analysis of genes from the predicted miR-TF network revealed a PPIN with three clusters, mainly involved in cellular biosynthesis, IL-1 signaling pathway, IFN-I production, cell-death regulation through apoptosis, pattern recognition, TLR signaling pathway, immune cell response, and importantly, Wnt and NF-κB signaling regulation (Figure 6. Further details are provided in Appendix A; see Table 5 for network properties). Local network clusters demonstrated the network implication in the canonical NF-κB pathway (FDR = 4.45 × 10^−8^). Thus, this PPIN retrieved from predicted elements of the miR-TF network illuminated their significance in immune cell receptors and inflammatory processes, mainly via the NF-κB pathway.

## 3. Discussion

Significant roles for miRs have been demonstrated in several human diseases, including autoimmune disorders [75]. MiRs are tremendously important in regulating immune system activity, and their dysfunction is closely associated with autoimmune diseases [76,77,78]. Generally, miRs regulate gene expression by forming RISC and subsequent binding to the target mRNA through specific sequences, which defines the binding affinity [79]. Although miR nucleotide sequences are highly conserved in mammals, polymorphisms in binding sites, or other sites related to miR structure and stability, often dramatically influence its function [79]. Thus, certain miR polymorphisms can be used as biomarkers for specific pathological conditions, including AITDs [17].

In the present study, we explored the association between miR-499 rs3746444 and HT risk in a selected study population from the southeast of Iran. Our results indicated that the rs3746444 polymorphism significantly increases HT risk in heterozygous, homozygous, dominant, overdominant, and allelic models. We also computed the potential targets and interactions of miR-499 using online databases. The results revealed that miR-499 critically participates in regulatory networks engaged in different facets of immune function, notably antigen recognition and processing, complement activity, cytokine production, inflammation, and immune receptor signaling.

MiR-499 can modulate inflammatory processes by regulating inflammatory cytokines and their receptors (e.g., IL-2, IL-6, IL-23a, IL-2RB, IL-8R, and IL-17RB), probably through NF-κB and TLR pathways [80,81]. This is consistent with our results from the predicted network that miR-499 may be essential for NF-κB and TLR signaling cascade regulation, thus regulating inflammatory conditions in HT. In addition, miR-499 regulates class II human leukocytic antigen (HLA-II), including HLA-DRB1, which is significantly associated with rheumatoid arthritis (RA), an autoimmune disease characterized by substantial chronic inflammatory condition [82]. Moreover, miR-499 exhibits anti-apoptotic activity by suppressing the pro-apoptotic proteins calcineurin A (CnA) α and CnAβ [83]. In harmony, our predicted regulatory miR-TF networks indicated that miR-499 is significantly associated with apoptosis and cell death regulation (Figure 4 and Figure 6).

The rs3746444 A/G polymorphism is located at the pre-miR area of miR-499 [84] and changes the A-U to G-U pairs; the latter is unstable in the pre-miR-499 stem, which can subsequently affect miR-499 maturation, leading to a lower level of miR-499-5p (a dominant miR-499 mature form) compared to the A allele [16]. An A-to-G substitution can potentially reduce the interacting ability of miRs with the 3’UTR of target genes, influencing their regulatory activity on gene expression or even changing their target [16,85]. Ding et al. showed that rs3746444, by affecting miR-499 maturation and decreasing miR-499-5p levels, reduces target (CnAα and CnAβ) suppression by miR-499, thereby decreasing its anti-apoptotic activity [16]. However, the definite effects observed by Ding and their colleagues can be changed in a different population due to other polymorphisms in upstream and downstream pathways or even target genes.

The correlation between miR-499 rs3746444 and predisposition to several autoimmune diseases has been investigated in numerous studies [78,86,87,88,89,90]. Hashemi et al. reported that rs3746444 was associated with an increased RA risk in an Iranian study population and that allele C was present more frequently in RA patients than in control subjects [78]. Conversely, this association was not observed in studies on the Chinese population [86,91,92]. Cai et al. reported that rs3746444 was positively associated with susceptibility to AITDs [17]. Their stratified analysis disclosed that the rs3746444 variant significantly increased the risk of GD but not HT. In a Mexican subpopulation, this polymorphism was associated with systemic lupus erythematosus (SLE) risk but not with GD or RA [84]. Other work showed that the C allele and CC and TC genotypes of rs3746444 significantly increased RA risk in a Mediterranean cohort [87], whereas Bin Yang and colleagues could not find any association between rs3746444 and RA risk [86]. Herein, we found that this polymorphism of miR-499 is associated with an increased risk of HT, and the C allele was markedly more prevalent in patients than controls.

To find potential contributors regarding the observed effect of miR-499 polymorphism, we predicted miR-499 target genes to construct PPIN, co-expression genes, and miR-TF networks. Intact protein modifications and PPINs are responsible for physiologically integrated and organized cellular functions, and perturbations could easily lead to pathological conditions and diseases. We predicted the PPIN specifically in thyroid tissue to narrow down the possible interactions between proteins and miR-proteins. The PPIN map is dynamic and depends on cell type and the time-point of expressing proteins [93]. Further, we predicted co-expression genes in thyroid tissue and transcription factors potentially interacting with miR-499 to regulate immune responses in thyroid and HT. The hub-driver nodes in these networks were identified as an integral component of immune system regulation. They control critical pathways in immunorecognition, immune tolerance, and the immune response to a stimulus. These results provide helpful information for future studies on identifying disease markers and developing novel therapeutic strategies for AITDs.

Aiming to verify the rationale of our predicted networks, we performed post-analyses based on existing evidence. The results revealed that the predicted elements are essential effectors of pathways pivotally engaged in immune system activities, potentially inducing chronic inflammation and autoimmune condition. Specifically, the core of enriched pathways is NF-κB and their regulators, including IκB, homeobox proteins such as Meox2 and Hox5a, and UBE2D3. Although its functions are ubiquitous and broad, the NF-κB is the major transcription factor for inflammatory processes and innate and adaptive immune response [94]. Any alteration in the homeostasis of NF-κB regulators can promote its activity, and therefore initiate an aberrant immune response [94]. The role of NF-κB in regulating apoptosis, remarkably immune system activity, and its association has been proposed for investigating the underlying pathogenesis of AITD [43]. Based on previous evidence and our findings, we suggest the miR-499 as an essential regulator of NF-κB in unbalanced immune response and chronic inflammation, seen in HT. Though not simple, one can assume that there could be an increased expression or function of NF-κB in patients carrying miR-499 rs3746444. However, this should be considered a potential starting point for future investigations in these patients, and may open a window for creating personalized treatment strategies.

On the other hand, there is a tendency to regulate immune receptor signalings, such as B and T cell receptors, TLR, and TRAF-mediated signaling pathways in our predicted networks. Several predicted network elements participate in antigene processing, mainly via ubiquitination and proteasome degradation, cell surface and membrane activity, interleukin signaling, and TRAF-mediated TLR cascades. These activities are crucial for immunorecognition and immune tolerance, and their dysregulation is potentially associated with an autoimmune reaction. Thus, we can assume that there could be a connection between miR-499 and immunorecognition mediated by TLR, BCR, and TCR signaling cascade and, importantly, NF-κB as a key player. These suggestions should be investigated with molecular assays of in vitro and in vivo studies.

Our sample size provides good statistical power (minimum statistical power of 0.8); however, further studies in different populations and ethnicities are needed to understand better the role of rs3746444 polymorphism in susceptibility to HT. Moreover, we could not rule out AITDs, including HT, in first-degree relatives of the control group by laboratory data since this study was not done prospectively. Furthermore, we did not assess the correlation between genotypes, allele frequency, and thyroid function, considering all cases were previously established and on medication with different individual treatment plans. Another limitation of the present work is choosing only one variant of the miR-499 for investigations, knowing that other possible functional polymorphisms in the loci can change the susceptibility to HT. Moreover, many miRs contribute to biological interaction that may lead to HT pathogenesis.

We could not assess the pathways and responsible genes experimentally, which were predicted by bioinformatics; these findings must be confirmed with in vitro and in vivo studies. The network analysis results could be confirmed using proteomics analysis, Western blot of the proteins, and cytokine assays on patient samples to confirm the NF-KB pathway. Nevertheless, our results showed factors that are critically involved in immune system regulation, pathways potentially associated with inflammation, and suggest, such as TRAF6-mediated signaling pathways and autoimmune conditions, NF-κB as a critical element interacting with miR-499. These findings provide helpful insight into possible underlying pathways and elements in HT etiopathogenesis. Although computational prediction may have limited value, it may help the study of targeted, focusing on a specific tissue and/or disease.

In summary, the present study suggests a significant correlation between the miR-499 rs3746444 polymorphism and HT susceptibility in an Iranian study population. Therefore, we propose that miR-499 rs3746444could be considered a prognostic biomarker to identify individuals susceptible to HT. Finally, we predicted interacting proteins and gene networks related to the miR; these findings will facilitate the design of tissue and target-specific studies to develop therapeutic strategies to treat AITDs.

## 4. Materials and Methods

### 4.1. Study Population

#### 4.1.1. Sample Size Calculation

The prevalence of HT was previously estimated in our population (1.5–3%) in the adult population [95]. Considering minor allele frequency (MAF) of 0.2, we calculated the sample size by the Cochran formula to have standard and reliable statistical power (minimum of 0.8) for this case-control study. Accordingly, the calculated sample size was 60 for the case group and 60 for the control group (ratio 1:1). However, we recruited all eligible HT patients (150 cases and 152 control) with medical records since our center is one of the referral endocrinology centers in southeast Iran.

#### 4.1.2. Patients and Controls

For this case-control study, we recruited a total of 302 subjects: 150 non-related HT patients and 152 non-related, ethnically matched healthy control subjects. This sample size was more than twice the calculated sample size. All HT cases were enrolled from patients referred to the endocrinology clinic, Ali-ebne Abitaleb Hospital, Zahedan University of Medical Sciences, Zahedan, Iran. Clinical manifestations and laboratory tests confirmed HT diagnosis. Confirmed HT patients had a different duration of the disease with various clinical characteristics and treatment regimens. All other overlapping thyroid diseases that may overlap with HT had been ruled out, and patients with other thyroiditis were excluded. Patients with other autoimmune diseases were excluded from the study. Matched healthy controls exhibiting normal thyroid function without any sign of goiter were recruited from participants in the check-up program of the Health Check-Up Center of the hospital. Control subjects suffering from or with a family history of autoimmune diseases (including AITD) were excluded. First- and second-degree relatives of all subjects were excluded from the study to maintain sample heterogeneity. The Ethics Committee of the National Institute for Medical Research Development (NIMAD) approved the present study (No. 958382); informed consent was obtained from all participants.

### 4.2. DNA Extraction and Genotyping

A 2 mL peripheral venous blood sample was drawn from all subjects by venipuncture, collected in an EDTA tube, and stored at −20 °C. Genomic DNA was extracted using the salting-out method. Genotyping of miR-499 rs3746444 polymorphism was performed by the PCR-RFLP method [96]; the primer sequences are shown in Table 5. PCR was carried out using commercially available Prime Taq premix (Genetbio, Nonsan, South Korea). Into every 0.20 mL PCR tube, 1 μL of genomic DNA (~100 ng/mL), 1 μL of each primer (10 μM), and 10 μL of 2 × Prime Taq Premix and 7 μL ddH2O were added. For both polymorphisms, the PCR thermocycler (Bio-Rad Laboratories Inc., Hercules, CA, USA) settings were as follows: 95 °C for 5 min, 30 cycles of 95 °C for 30 s, 64 °C for 30 s, 72 °C for 30 s, and a final extension of 72 °C for 10 min. Ten microliters of PCR product were digested by the appropriate restriction enzymes (Table 6) and separated by electrophoresis on an agarose gel; representative photographs are shown in Figure 7. Approximately 20% of random samples were re-genotyped; the results confirmed previous genotyping outcomes.

### 4.3. Prediction of Tissue-Specific, Co-Expression, and miR-TF Gene Networks

Prediction of the target genomic sites of miR-499 rs3746444 (T > C) was made by TargetScanHuman (v.7.2) using its default setting [10]. To find the sub-regulatory networks related to these genomic variations, network data for (1) affected tissue-specific protein–protein interaction networks (PPIN) were retrieved from the DifferentialNet database (Available at http://netbio.bgu.ac.il/diffnet; accessed on 28 July 2021) [97], (2) the tissue-specific co-expression genes were retrieved from the TCSBN database [98], and (3) miR-TF-miR co-regulatory interactions were retrieved from RegNetwork (Available at http://regnetworkweb.org; accessed on 29 July 2021) [99]. All networks were constructed by the top 100 predicted target genes of miR-499 with the lowest context score (CS). The CS was calculated for each specific site of the miR target as the sum of the contribution of 14 features introduced in Agarwal et al. (2015) [100]; the regulatory modules were constructed by Networkanalyst (v. 3.0) [101]. Functional regulatory clusters of affected networks were enriched by the Reactome biological pathway database [102].

### 4.4. Post-Analysis of Predicted Networks

We performed post-analysis using the STRING database (v.11.5) [103] to verify our predicted networks based on existing evidence. Hence, we selected the top predicted genes based on their degree and betweenness (Appendix A) to verify that predicted networks and their core elements significantly contribute to enriched pathways. The networks properties are provided in Table 5 (also see Appendix A for cluster detail). This was done to compensate for the lack of verifying molecular and biological assays in this study, providing a high level of evidence for our results.

### 4.5. Statistical Analysis

Statistical analysis was performed using SPSS software (version 22). The χ2 test and independent-sample *t*-test were applied for categorical and continuous data, respectively. Allele and genotype frequency distributions of the variants in patients and control subjects were expressed as percentages of the total number of alleles and genotypes. Odds ratios (ORs) and 95% confidence intervals (95% CIs) were calculated by unconditional logistic regression analysis. Differences with a *p* < 0.05 were considered statistically significant. For both polymorphisms, deviation from the Hardy–Weinberg equilibrium (HWE) was assessed using the χ2 test.

## Figures and Tables

**Figure 1 ijms-22-10094-f001:**
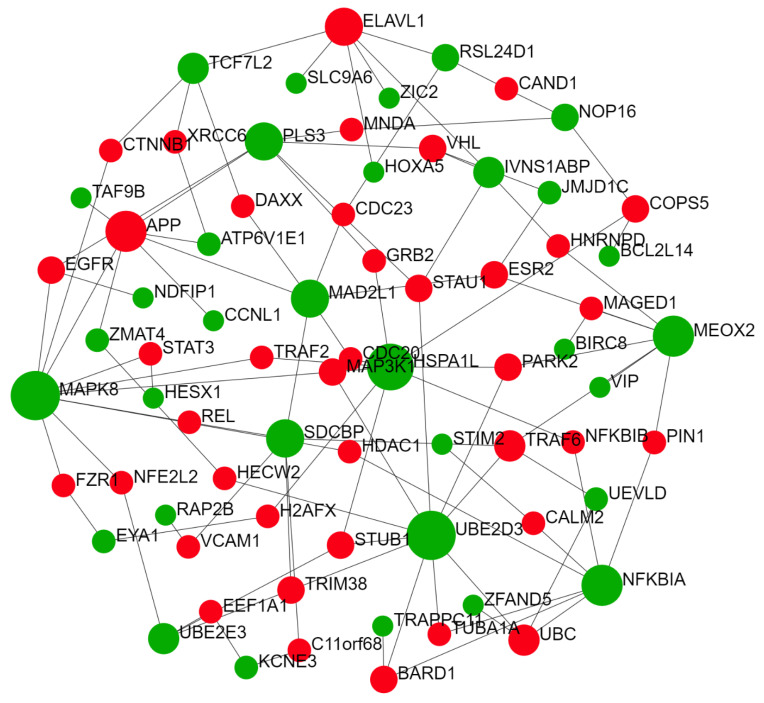
Tissue-specific protein–protein interaction network related to miR-499 functional modules. This network revealed *MEOX2*, *SDCBP*, *UBE2D3*, *MAPK8*, and *NFKBIA* as main hub-driver nodes (green nodes) in thyroid tissue (for statistical network details, see Appendix A). These genes are mainly implicated in adaptive and innate immune system antigen processing and pattern recognition, mainly throughout toll-like receptors (TLRs) signaling. miR, microRNA; MEOX2, mesenchyme homeobox 2; SDCBP, syndecan binding protein; UBE2D3, ubiquitin-conjugating enzyme E2 D3; MAPK8, mitogen-activated protein kinase 8; NFKBIA, nuclear factor-kappa B inhibitor alpha.

**Figure 2 ijms-22-10094-f002:**
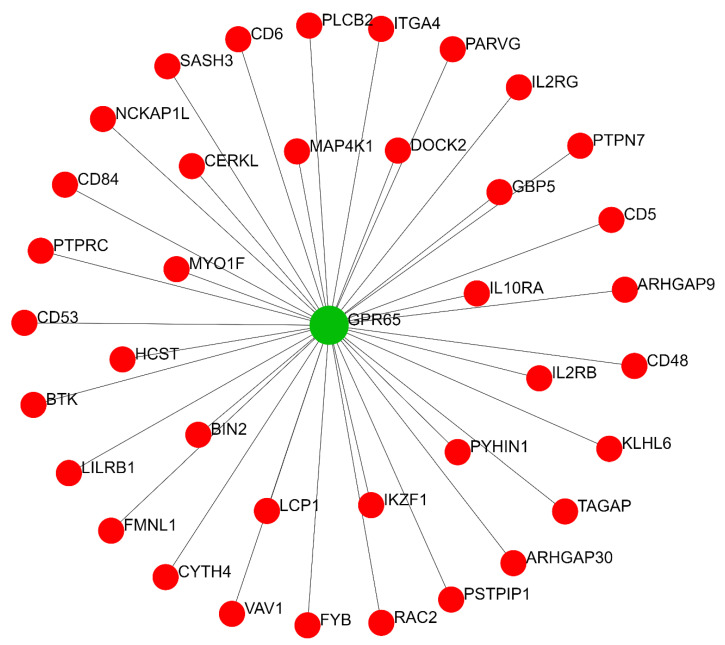
Co-expression gene regulatory network related to miR-499 functional modules. *GPR65* is the seed node (illustrated in green). Peripheral red nodes indicate co-expressed nodes (the statistical network details are outlined in Appendix A). miR, microRNA; GPR65, G protein-coupled receptor 65.

**Figure 3 ijms-22-10094-f003:**
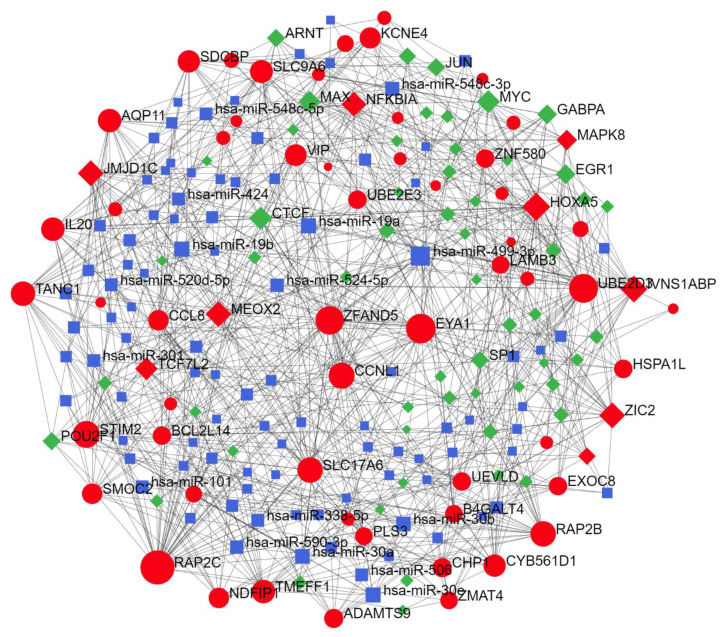
miR-TF regulatory network related to miR-499 functional modules in thyroid tissue. *ZIC2*, *NFKBIA*, *RAP2C*, *HOXA5*, and *TCF7L2* were identified as the main driver nodes. Red nodes indicate seed nodes used for network reconstruction (see Appendix A for the statistical network details), green nodes indicate TFs, and blue nodes indicate miRs. miR, microRNA; TF, transcription factor; ZIC2, zic family member 2; NFKBIA, nuclear factor-kappa B inhibitor alpha; RAP2C, member of RAS oncogene family; HOXA5, homeobox A5; TCF7L2, transcription factor 7 like 2.

**Figure 4 ijms-22-10094-f004:**
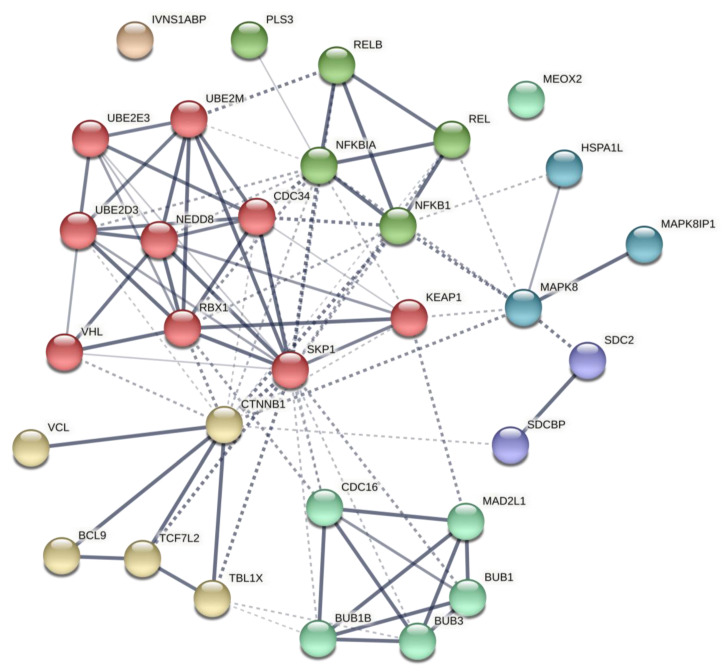
Network 1: Interaction network based on initial predicted PPIN. This network was reconstructed by 12 genes with the highest degree of predicted regulatory miR-499 PPIN as inputs. Each cluster is illustrated with a different color. Edges represent protein–protein associations. Chromatic edges indicate highest (0.9), high (0.7), medium (0.4), and low (0.15) confidence. Inter-cluster edges are shown with dashed lines. PPIN, protein-protein interaction network; miR, microRNA.

**Figure 5 ijms-22-10094-f005:**
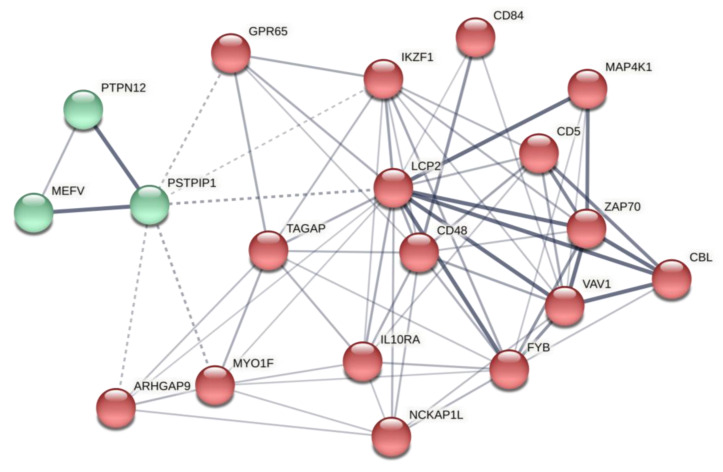
Network 2: Interaction network based on initial predicted co-expression regulatory network. This network was reconstructed by 14 genes involved in the predicted co-expression regulatory network as inputs. Each cluster is illustrated with a different color. Edges represent protein–protein associations. Chromatic edges indicate highest (0.9), high (0.7), medium (0.4), and low (0.15) confidence. Inter-cluster edges are shown with dashed lines.

**Figure 6 ijms-22-10094-f006:**
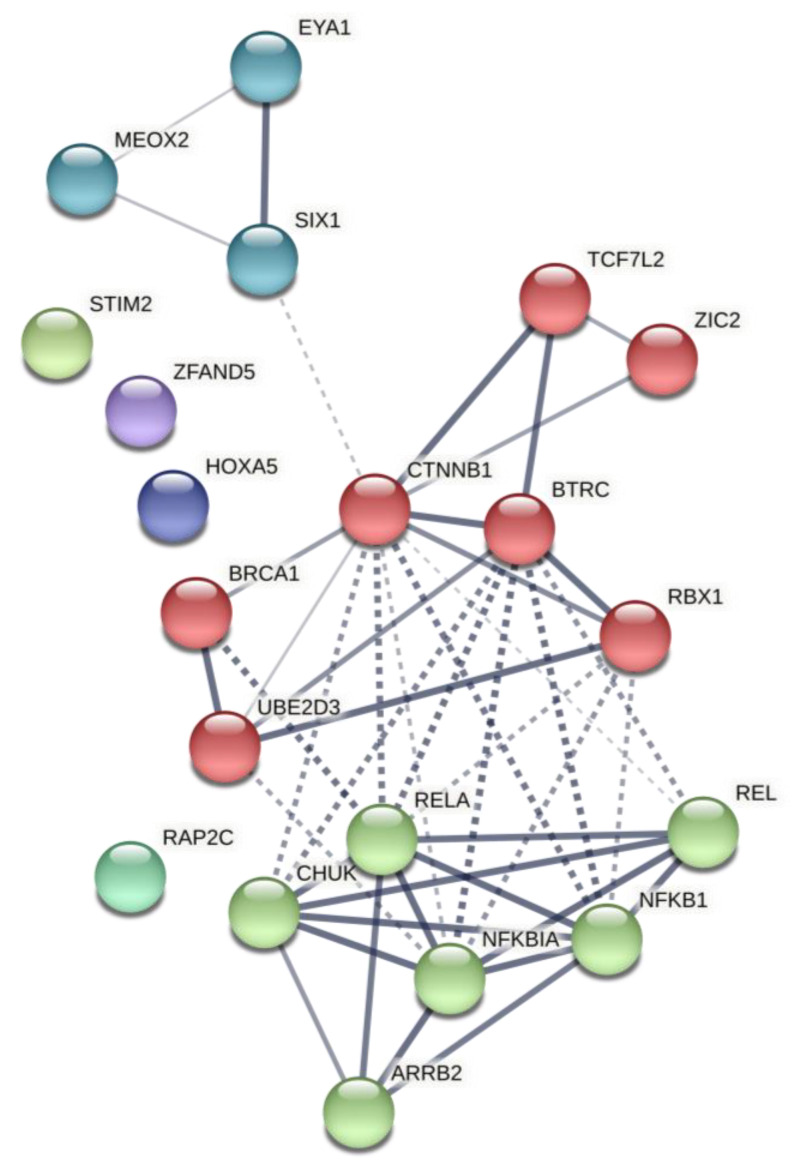
Network 3: Interaction network based on initial predicted miR-TF network. This network was reconstructed by ten genes with the highest degree involved in the predicted regulatory miR-TF networks. Each cluster is illustrated with a different color. Edges represent protein–protein associations. Chromatic edges indicate highest (0.9), high (0.7), medium (0.4), and low (0.15) confidence. Inter-cluster edges are shown with dashed lines. miR, microRNA; TF, transcription factor.

**Figure 7 ijms-22-10094-f007:**
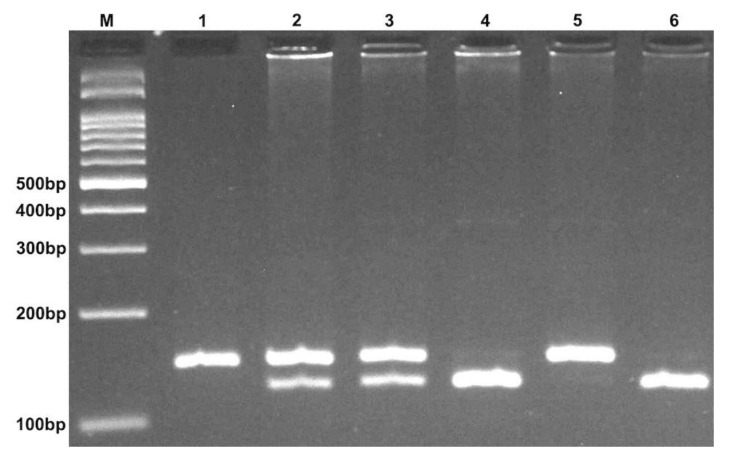
Electrophoresis pattern of the PCR-RFLP method for the detection of miR-499 rs3746444 polymorphism. M: DNA marker; Lanes 1,5: CC; Lanes 2,3: CT; Lanes 4,6: TT.

**Table 1 ijms-22-10094-t001:** Demographic status of the study subjects.

Demographic		HT, *n* = 150	Control, *n* = 152	*p*-Value
Age, year (mean ± SD)		37.96 ± 12.06	36.22 ± 12.86	0.226
Sex	Male	17	24	0.314
	Female	133	128	

Abbreviations: HT, Hashimoto thyroiditis; SD, standard deviation.

**Table 2 ijms-22-10094-t002:** The genotype and allele frequencies of miR-499 rs3746444 T > C polymorphism in Hashimoto thyroiditis (HT) patients and control subjects.

Polymorphisms	Gentic Model	Genotype	HT, *n* (%)	Controls, *n* (%)	OR (95% CI)	*p*-Value
rs3746444 (miR-499)	Codominant	TT	37 (24.7)	78 (51.3)	1.00	-
		CT	93 (62.0)	59 (38.8)	3.32 (2.00–5.53)	<0.001
		CC	20 (13.3)	15 (9.9)	2.81 (1.30–6.10)	0.014
	Dominant	TT	37 (24.7)	78 (51.3)	1.00	-
		CT + CC	113 (75.3)	74 (48.7)	3.22 (1.97–5.25)	<0.001
	Recessive	TT + CT	130 (86.7)	137 (90.1)	1.00	-
		CC	20 (13.3)	15 (9.9)	1.41 (0.69–2.86)	0.447
	Overdominant	CC + TT	57 (38)	93 (61.2)	1.0	-
		CT	93 (62.0)	59 (38.8)	2.57 (1.62–4.09)	<0.001
	Allele	T	167 (55.7)	215 (70.7)	1.00	-
		C	133 (44.3)	89 (29.3)	1.92 (1.37–2.69)	0.001

Abbreviations: HT, Hashimoto thyroiditis; miR, microRNA; OR, odds ratio; CI, confidence interval.

**Table 3 ijms-22-10094-t003:** Top-ranked hub nodes of constructed networks.

Networks	Label	Degree	Betweenness
Tissue specific PPIN	*MEOX2*	174	105,725.28
	*SDCBP*	83	62,693.4
	*UBE2D3*	71	58,615.64
	*MAPK8*	71	52,068.96
	*NFKBIA*	47	29,486.37
Co-expression gene network	*GPR65*	37	666
	*TAGAP*	1	0
	*MYO1F*	1	0
	*MAP4K1*	1	0
	*CD5*	1	0
miR-TF network	*ZIC2*	292	220,452.09
	*NFKBIA*	102	71,452.21
	*RAP2C*	76	29,506.18
	*HOXA5*	72	33,168.41
	*TCF7L2*	58	33,137.3

Abbreviations: MEOX2, mesenchyme homeobox 2; SDCBP, syndecan binding protein; UBE2D3, ubiquitin-conjugating enzyme E2 D3; MAPK8, mitogen-activated protein kinase 8; NFKBIA, nuclear factor-kappa B inhibitor alpha; GPR65, G protein-coupled receptor 65; TAGAP, T-cell activation RhoGTPase activating protein; MYO1F, myosin IF; MAP4K1, mitogen-activated protein kinase kinase kinase kinase 1; CD5, CD5 molecule; ZIC2, zic family member 2; RAP2C, member of RAS oncogene family; HOXA5, homeobox A5; TCF7L2, transcription factor 7 like 2; miR-TF, microRNA-transcription factors; PPIN, protein–protein interaction network.

**Table 4 ijms-22-10094-t004:** Top enriched pathways related to tissue-specific functional sub-networks of miR-499. Each section depicts the top 10 pathways for each network. The upper section shows pathways enriched by the tissue-specific PPIN, mostly involved in antigen recognition and signaling cascade. The middle section displays pathways enriched by the tissue-specific gene co-expression network, of which the main driver nodes are involved in Rho GTPase signaling, cytokine signaling, and adaptive immune system regulation. The bottom section lists the pathways enriched by miR-TF in thyroid tissue, indicating its role in immune system signaling (mainly via toll-like receptors) cascades and inflammation induction.

Pathway Enriched by Tissue-Specific Protein–Protein Interaction Network					
Pathway	Total	Expected	Hits	*p*-Value	FDR *
Antigen processing: ubiquitination and proteasome degradation	224	10.4	50	9.93 × 10^−22^	1.39 × 10^−18^
Immune system	1140	52.7	120	6.66 × 10^−21^	4.67 × 10^−18^
Class I MHC mediated antigen processing and presentation	267	12.4	53	1.66 × 10^−20^	7.77 × 10^−18^
Adaptive immune system	654	30.3	83	9.72 × 10^−19^	3.41 × 10^−16^
TRIF-mediated TLR3/TLR4 signaling	87	4.03	23	3.91 × 10^−12^	1.02 × 10^−09^
MyD88-independent cascade	88	4.07	23	5.07 × 10^−12^	1.02 × 10^−9^
TLR3 cascade	88	4.07	23	5.07 × 10^−12^	1.02 × 10^−9^
Activated TLR4 signalling	100	4.63	24	1.26 × 10^−11^	2.21 × 10^−9^
TLR4 cascade	103	4.77	24	2.49 × 10^−11^	3.87 × 10^−9^
RIG-I/MDA5 mediated induction of IFN-alpha/beta pathways	67	3.1	19	8.12 × 10^−11^	1.14 × 10^−8^
**Pathway Enriched by Gene Co-Expression Network**					
Rho GTPase cycle	123	0.368	5	2.43 × 10^−5^	1.70 × 10^−2^
Signaling by Rho GTPases	123	0.368	5	2.43 × 10^−5^	1.70 × 10^−2^
Immune system	1140	3.41	11	1.26 × 10^−4^	0.0589
Interleukin−3, 5 and GM-CSF signaling	51	0.153	3	4.37 × 10^−4^	0.153
Immunoregulatory interactions between a lymphoid and a non-lymphoid cell	80	0.24	3	1.63 × 10^−3^	0.457
Adaptive immune system	654	1.96	7	2.08 × 10^−3^	0.479
Hemostasis	511	1.53	6	2.99 × 10^−3^	0.479
Cell surface interactions at the vascular wall	99	0.297	3	0.003	0.479
Interleukin receptor SHC signaling	28	0.0839	2	0.00308	0.479
Antigen activates B-cell receptor leading to generation of second messengers	32	0.0959	2	0.00401	0.506
**Pathway Enriched by miR-TF Network**					
TRIF-mediated TLR3/TLR4 signaling	87	3.15	25	1.65 × 10^−16^	1.04 × 10^−13^
MyD88-independent cascade	88	3.19	25	2.23 × 10^−16^	1.04 × 10^−13^
TLR3 cascade	88	3.19	25	2.23 × 10^−16^	1.04 × 10^−13^
Activated TLR4 signalling	100	3.62	26	5.98 × 10^−16^	2.09 × 10^−13^
TLR4 cascade	103	3.73	26	1.31 × 10^−15^	3.68 × 10^−13^
TRAF6 mediated induction of proinflammatory cytokines	62	2.25	20	1.66 × 10^−14^	3.89 × 10^−12^
TLR10 cascade	74	2.68	21	6.41 × 10^−14^	9.99 × 10^−12^
TLR5 cascade	74	2.68	21	6.41 × 10^−14^	9.99 × 10^−12^
MyD88 cascade initiated on plasma membrane	74	2.68	21	6.41 × 10^−14^	9.99 × 10^−12^
TRAF6 mediated induction of NFkB and MAP kinases upon TLR7/8 or 9 activation	76	2.75	21	1.15 × 10^−13^	1.62 × 10^−11^

* adjusted *p*-value for multiple testing by Benjamini–Hochberg method. Abbreviations: FCGR, Fc gamma receptor; FDR, false discovery counts; MyD88, myeloid differentiation primary response 88; TLR, toll-like receptor; TRAF6, TNF receptor-associated factor 6.

**Table 5 ijms-22-10094-t005:** Network properties and statistics are presented.

Description	Nodes	Edges	Expected Number of Edges	Avg. Node Degree	Avg. local Clustering Coefficient	Inflation Parameter (MCL)	Enrichment *p*-Value
Network 1 (based on initial PPIN)	31	96	46	6.19	0.627	4	1.24 × 10^−10^
Network 2 (based on initial miR-TF)	20	45	23	4.5	0.612	4	3.16 × 10^−5^
Network 3 (based on initial co-expression)	19	68	8	7.16	0.718	4	<1.0 × 10^−16^

**Table 6 ijms-22-10094-t006:** Primer sequences were used for the detection of miR-499 rs3746444 T > C by the PCR-RFLP method.

Polymorphisms	Primer Sequence (5′→3′)	Restriction Enzyme	Fragment (bp)
rs3746444 T > C (miR-499)	F: CAAAGTCTTCACTTCCCTGCCA R: GATGTTTAACTCCTCTCCACGTGATC	BclI	CC: 146 CT: 146 + 122 + 24 TT: 122 + 24

Abbreviations: miR, microRNA, bp, base pair.

## Data Availability

The data presented in this study are available on request from the corresponding author.

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
