# Peer review of "Association of miR-499 Polymorphism and Its Regulatory Networks with Hashimoto Thyroiditis Susceptibility: A Population-Based Case-Control Study"

_ijms, 2021, doi:10.3390/ijms221810094_

Round 1

Reviewer 1 Report

The issues have been handled accordingly.

Author Response

Thanks for the reviewer’s encouraging feedback which has significantly improved our manuscript

Reviewer 2 Report

The corrections made are reasonably adequate for the publication of the
article. I have nothing else to add.

Author Response

Thanks for the reviewer’s encouraging feedback which has significantly improved our manuscript.

This manuscript is a resubmission of an earlier submission. The following is a list of the peer review reports and author responses from that submission.

Round 1

Reviewer 1 Report

The paper by Tabasi et al is a genetic case-control study of Hashimoto thyroiditis. Of the studied SNPs in miR genes, one shows association the other does not. I think that the latter one should not be included in the subsequent analysis (i.e. figures, table) as it is probably not relevant in the pathogenesis. Meaning of some parameters (i.e. betweenness) need also be expalined.

Author Response

Comment: The paper by Tabasi et al is a genetic case-control study of Hashimoto thyroiditis. Of the studied SNs in miR genes, one shows association the other does not.

Point 1: I think that the latter one should not be included in the subsequent analysis (i.e. figures, table) as it is probably not relevant in the pathogenesis.

Response: We appreciate the reviewer’s comments. The purpose behind reported results of the microRNA-125a (miR-125a), which showed no significant association with Hashimoto thyroiditis (HT) risk, is reporting negative results of a polymorphism in a certain population. There are two main reasons which we briefly discuss in the following sentences.  A), these polymorphisms are varied in different populations. Although this polymorphism showed no significant association, it may have an association with HT in different populations and ethnic groups. This is the point of matter for personalized diagnosis and treatment. If we do not report the negative result, further studies, particularly meta-analysis, may have biased data with inconclusive results. On the other hand, numerous studies reported negative results regarding a given polymorphism (Miyamoto-Mikami et al., 2019 [PMID: 31791263], Xiao et al., 2019 [PMID: 31804315], Heemstra et al., 2009 [PMID: 19018782], Penna-Martinez et al., 2009 [PMID: 19961590], Bienkowski et al., 2015 [PMID: 25933951], Li et al., 2013 [PMID: 24129496], Spurdle et al., 2002 [PMID: 12142725], Peterlin et al., [PMID: 29876231], Artukovic et al., 2020 [PMID: 33410302], Zhang et al., 2020 [PMID: 32493081]). B), our sample size may appear enough for such studies, but these results should not be interpreted as definitive results. Although our study has enough statistical power to reach a reasonable conclusion (consider the relatively low prevalence of HT in the general population), the larger sample size from this population may show a different result (this is a possibility). So, these are logics that we reported the negative result of miR-125a. Some studies reported an association between this miR and AITD, which shows an ethnic/population-related trend (Inoue et al., 2014 [PMID: 24990808], Liu et al., 2-2- [PMID: 32595646], Cai et al., 2017 [PMID: 27888002]).

Comment: Point 2: Meaning of some parameters (i.e. betweenness) need also be explained.

Response: Thanks for the remark. We added the explanation of degree and betweenness, and total, expected, and hits terms of enrichment analysis, in the second part of the Results section (page 4, lines 138-143 and lines 146-151). Also, we added the description of FDR in Table 4 (page 8, lines 182 at the footnote of Table 4).

Reviewer 2 Report

The work has been very well conducted and can represent the starting
point for further studies in the same direction. An important limitation
is the small number of samples and the selection of a single one.
These limits have been correctly reported in the discussion, however
the choice of population may influence some of the results for the known
relationships to the thyroid gland conformation, nutrition and influence
of environmental factors. This section should be better explained in
the discussion while not avoiding the reliable results. A better description
of the selection of the population with related clinical and morphological
thyroid characteristics could have helped the interpretation of the results,
especially in light of the wide heterogeneity of the thyroid glandular response
related to the morphological trend.

Author Response

The work has been very well conducted and can represent the starting point for further studies in the same direction.

Comment: An important limitation is the small number of samples and the selection of a single one. These limits have been correctly reported in the discussion, however the choice of population may influence some of the results for the known relationships to the thyroid gland conformation, nutrition and influence of environmental factors. This section should be better explained in the discussion while not avoiding the reliable results.

Response: The authors acknowledge the wise advice and vision of the respected reviewer. We did not consider factors other than genetic susceptibility, knowing a high probability of genetic predisposition for Hashimoto thyroiditis (HT). This study just explored the connection between the polymorphisms and HT, and we had to recruit all patients with confirmed Hashimoto thyroiditis diagnosis, not those who may have overlapped signs/symptoms in the differential diagnosis. Thus, all patients were selected based on their previous medical records, and their disease was confirmed clinically and by laboratory assessments. None of them were new patients. On the other hand, given that HT is not a very common disease (0.5-1.5 cases/1000 person/year) compared to the general population, we had the problem for collecting a larger sample size with controlled potential contributing/confounding factors. We have done the power analysis to make sure the statistical power was enough to produce reliable results. We added an explanation in this regard in the Discussion, lines 357-361.

Comment: A better description of the selection of the population with related clinical and morphological thyroid characteristics could have helped the interpretation of the results, especially in light of the wide heterogeneity of the thyroid glandular response related to the morphological trend.

Response: Thanks for the thoughtful comment. This is correct, and we are aware of this limitation. In the initial survey, we tried to report thyroid features and function tests and other clinical features in these patients. However, enrolled patients had different disease duration and treatment plans, and different clinical characteristics. Thus, we could not find a large sample size for each category (such as newly diagnosed case, same treatment regimen, and initial presentation), considering that HT is not a common disease in general. This study was a part of a notational regional program that assess the relationship between genetic polymorphisms and particular autoimmune diseases and cancers in the southeast of Iran, and this study’s results could be interpreted as preliminary results, as the reviewer wisely mentioned, a starting point for further studies.

We added more details regarding patient enrollment in this study in the Materials and Methods, lines 384-387.

Round 2

Reviewer 1 Report

All figures still contain the irrelevant miR as significant contributor.